# Downregulation of miR-122-5p Activates Glycolysis via PKM2 in Kupffer Cells of Rat and Mouse Models of Non-Alcoholic Steatohepatitis

**DOI:** 10.3390/ijms23095230

**Published:** 2022-05-07

**Authors:** Yosuke Inomata, Jae-Won Oh, Kohei Taniguchi, Nobuhiko Sugito, Nao Kawaguchi, Fumitoshi Hirokawa, Sang-Woong Lee, Yukihiro Akao, Shinji Takai, Kwang-Pyo Kim, Kazuhisa Uchiyama

**Affiliations:** 1Department of General and Gastroenterological Surgery, Osaka Medical and Pharmaceutical University, 2-7 Daigaku-machi, Takatsuki, Osaka 569-8686, Japan; yosuke.inomata@ompu.ac.jp (Y.I.); nao.kawaguchi@ompu.ac.jp (N.K.); fumitoshi.hirokawa@ompu.ac.jp (F.H.); sang-woong.lee@ompu.ac.jp (S.-W.L.); uchi@ompu.ac.jp (K.U.); 2Department of Applied Chemistry, Institute of Natural Science, Global Center for Pharmaceutical Ingredient Materials, Kyung Hee University, Yongin 17104, Korea; odinplate@naver.com (J.-W.O.); kimkp@khu.ac.kr (K.-P.K.); 3Translational Research Program, Osaka Medical and Pharmaceutical University, 2-7 Daigaku-machi, Takatsuki, Osaka 569-8686, Japan; 4United Graduate School of Drug Discovery and Medical Information Sciences, Gifu University, 1-1 Yanagido, Gifu, Gifu 501-1193, Japan; nsugito@gifu-u.ac.jp (N.S.); yakao@gifu-u.ac.jp (Y.A.); 5Department of Innovative Medicine, Graduate School of Medicine, Osaka Medical and Pharmaceutical University, 2-7 Daigaku-machi, Takatsuki, Osaka 569-8686, Japan; shinji.takai@ompu.ac.jp

**Keywords:** NASH, microRNA (miRNA), miR-122-5p, PKM2, Kupffer cells, macrophages, Warburg effect, glycolysis, hepatocellular carcinoma, phosphoproteomics

## Abstract

Non-alcoholic steatohepatitis (NASH) has pathological characteristics similar to those of alcoholic hepatitis, despite the absence of a drinking history. The greatest threat associated with NASH is its progression to cirrhosis and hepatocellular carcinoma. The pathophysiology of NASH is not fully understood to date. In this study, we investigated the pathophysiology of NASH from the perspective of glycolysis and the Warburg effect, with a particular focus on microRNA regulation in liver-specific macrophages, also known as Kupffer cells. We established NASH rat and mouse models and evaluated various parameters including the liver-to-body weight ratio, blood indexes, and histopathology. A quantitative phosphoproteomic analysis of the NASH rat model livers revealed the activation of glycolysis. Western blotting and immunohistochemistry results indicated that the expression of pyruvate kinase muscle 2 (PKM2), a rate-limiting enzyme of glycolysis, was upregulated in the liver tissues of both NASH models. Moreover, increases in PKM2 and p-PKM2 were observed in the early phase of NASH. These observations were partially induced by the downregulation of microRNA122-5p (miR-122-5p) and occurred particularly in the Kupffer cells. Our results suggest that the activation of glycolysis in Kupffer cells during NASH was partially induced by the upregulation of PKM2 via miR-122-5p suppression.

## 1. Introduction

Non-alcoholic fatty liver disease (NAFLD) is associated with a risk of its progression to non-alcoholic steatohepatitis (NASH). NASH has a similar histopathology to that of alcoholic hepatitis, with steatosis, inflammation, and hepatocellular ballooning in the liver, yet without a history of alcohol consumption [1,2]. Approximately 15% of NASH patients develop cirrhosis or hepatic decompensation within the same time period [3], and as many as 11% of NASH-associated cirrhosis cases progress to hepatocellular carcinoma (HCC) [4]. Therefore, preventing NASH progression is of major importance.

The Warburg effect is a cancer-specific adaptation of energy metabolism. In cancer cells, despite the aerobic conditions, the glycolytic pathway is activated instead of the tricarboxylic acid (TCA) cycle [5]. Pyruvate kinases catalyze the final step of glycolysis to yield pyruvate. These kinases are classified into four isoforms, namely PKL, PKR, PKM1, and pyruvate kinase muscle 2 (PKM2). Among these isoforms, PKM2 is a central regulator of the Warburg effect, particularly glycolysis [6].

We have previously reported that PKM2 is upregulated and related to cell growth not only in cancer tissue [7,8,9], but also in precancerous lesions such as colorectal adenomas [10,11]. Furthermore, PKM2 is upregulated in macrophages during inflammatory conditions such as sepsis, thereby enhancing macrophage activation [12,13]. Few studies have explored the relationship between NASH and the Warburg effect. In particular, the molecular mechanism of the Warburg effect in liver-specific Kupffer cells remains unknown.

MicroRNAs (miRNAs) are small non-coding RNAs that regulate gene expression at the post-transcriptional level [14]. MicroRNA122-5p (miR-122-5p) is a liver-specific miRNA comprising 70% and 52% of the total miRNA contents in the livers of adult mice and humans, respectively [15,16]. Although miR-122-5p expression is reported to be decreased in NASH patients and animal models [17,18], there are limited data on the dynamics of miR-122-5p in Kupffer cells during NASH, and the associated molecular mechanism remains largely unknown.

We have recently employed several new models, exploring the pathogenic mechanism of NASH and its relevant therapeutics [19,20]. In the current study, we investigate the Warburg effect in these NASH models, with a focus on Kupffer cells and miRNA regulation.

## 2. Results

### 2.1. HFC-Fed Rats Developed NASH after 8 Weeks

To investigate NASH pathophysiology, we used the previously generated appropriate rat models [21]. Ten-week-old SHRSP5/Dmcr rats were fed a standard diet (SD; control group) or a high-fat and high-cholesterol (HFC) diet for eight weeks (NASH group).

The livers of NASH group rats presented hepatomegaly and paler color upon macroscopic examination (Figure 1A). The liver/body weight ratio in the NASH group was higher than that in the control group **(**Figure 1B). Both aspartate transaminase (AST) and alanine transaminase (ALT) plasma levels in the NASH group were significantly higher than those in the control group (Figure 1C). Hematoxylin–eosin (HE) staining revealed severe steatosis, lipid droplets, and hepatocyte ballooning in the NASH group, closely resembling human NASH. Sirius Red staining indicated areas of fibrosis in the NASH group, consistent with areas of steatosis (Figure 1D). Liver specimens from the control and NASH groups were scored for steatosis, inflammation, ballooning, and NAFLD activity scores (NAS) [22] (Appendix A). These data were consistent with the data reported previously [21].

### 2.2. The Glycolysis Pathway Was Activated and PKM2 Was Upregulated in NASH Tissues

Next, we investigated the protein expression and phosphorylation as an index of activation. Phosphoproteomics using LC-MS/MS identified 3313 peptides (1392 proteins) and 1340 phosphopeptides with high confidence (PhosphoRS > 0.75; Appendix A). Comparative phosphoproteomic analysis between NASH and control samples revealed 302 differentially expressed proteins (cut-off: *p*-value < 0.05 or protein intensity fold change > 2, <0.5; Appendix A). The total protein expression heat map indicated that cell–cell adhesion, apoptotic process, immune response, gluconeogenesis, and RNA translation were significantly enriched in the NASH group. In contrast, the oxidation–reduction process, TCA cycle, and lipid metabolic process were considerably decreased in the NASH group (Figure 2A). We hypothesized that glycolysis was activated as the TCA cycle was suppressed in NASH. Both the protein and phosphorylation levels of PKM were considerably upregulated in the NASH group (Figure 2B and Appendix A). Next, we compared the hepatic levels of PKM2, phosphorylated PKM2 (p-PKM2), and pyruvate kinase L/R (PKLR) between the groups. Both PKM2 and p-PKM2 levels were increased in the NASH group. In contrast, the expression level of PKLR, an isozyme of PKM2 present in normal hepatocytes [23,24], was decreased in NASH (Figure 2C). Similar results were observed via immunohistochemistry (IHC) staining, and PKM2 loci were replaced by PKLR foci (Figure 2D). These results suggested that glycolysis was activated in our rat model.

### 2.3. CDAHFD-Fed Mice Developed NASH

We next confirmed our results in the NASH mouse model [25]. Six-week-old C57BL/6J-NASH mice were fed a SD (control group) or choline-deficient, L-amino-acid-defined, high-fat diet with 0.1% methionine (CDAHFD) for 1–12 weeks (NASH group). In the NASH group, CDAHFD for 12 weeks induced hepatomegaly and paler color upon macroscopic examination (Figure 3A). The liver/body weight ratio in the NASH group was higher than that in the control group (Figure 3B). Both AST and ALT plasma levels were also significantly higher (Figure 3C). HE staining revealed severe steatosis, lipid droplets, and hepatocyte ballooning in the NASH group. Sirius Red staining indicated fibrotic areas, consistent with areas of steatosis in NASH (Figure 3D). Liver specimens from the control and NASH groups were scored for steatosis, inflammation, ballooning, and NAS (Appendix A). Biochemical examination of blood and pathological changes were consistent with those reported previously [25]. Further, both PKM2 and p-PKM2 were upregulated. In contrast, hepatic PKLR expression was lower in the NASH group compared to controls (Figure 3E,F).

### 2.4. PKM2 Expression Was Upregulated in the Early Phase of NASH

To explore the molecular pathophysiology of NASH, we prepared early-phase NASH mouse models at 1, 3, and 9 weeks. HE and Sirius Red staining revealed lipid droplets and fibrosis in the initial phase of NASH (Figure 4B), while NAS increased with the number of weeks of feeding (Appendix A). Furthermore, PKM2 and p-PKM2 upregulation was detected at one week, with both increasing over time. In contrast, PKLR expression gradually decreased in parallel (Figure 4A,B). These results suggested that the Warburg effect is increasingly activated in NASH over time.

### 2.5. The miR-122-5p Targeted PKM2 Directly and Was Downregulated in Kupffer Cells during NASH

We next explored the mechanism of PKM2 upregulation with a focus on miRNA expression. Extensive analysis of the Target Scan7.2 database (http://www.targetscan.org, accessed on 31 March, 2022) indicated that miR-122-5p targets the 3′-untranslated region (3′UTR) of *PKM* (Table 1). To determine whether miR-122-5p directly targets PKM2, we used an HCC cell line (HuH-7). The expression levels of PKM2 and p-PKM2 were downregulated in the miR-122-5p-transfected HCC cell line. Treatment with an inhibitor of miR-122-5p reversed these expression changes (Figure 5A left). The luciferase reporter activity of wild-type PKM2 was significantly inhibited after the introduction of miR-122-5p into HuH-7 cells. In contrast, mutation of the *PKM* 3′UTR-binding site abolished the inhibitory ability of miR-122-5p (Figure 5B). The expression of miR-122-5p in the livers of NASH model subjects was significantly lower compared to that in the healthy livers of both animal models (Figure 5C).

A recent study indicated that the expression of PKM2 was upregulated in Crohn’s disease, which is an inflammatory condition [26]. Furthermore, inflammatory injury induces an increase in PKM2 in macrophages [27]. Thus, we assumed that PKM2 is upregulated in Kupffer cells during NASH. To determine the localization of PKM2, we performed IHC staining with antibodies against Kupffer cell markers CD68 for rats and F4/80 for mice. IHC staining revealed co-localization of PKM2 and Kupffer cells (Figure 5D). The expression of miR-122-5p in NASH group Kupffer cells was significantly lower compared to that in the control group (Figure 5E). Furthermore, the regulatory mechanism of PKM2 expression by miR-122-5p was also confirmed in the Kupffer cells (KUP5) (Figure 5A right). Taken together, these findings suggest that the downregulation of miR-122-5p enhanced PKM2 expression, thereby driving the Warburg effect in Kupffer cells during NASH (Figure 6).

## 3. Discussion

This study explored the relationship between NASH and the Warburg effect in animal models. PKM2, a central factor in glycolysis, was activated in NASH. In addition, we established that PKM2 was upregulated via the dysregulation of miR-122-5p in Kupffer cells.

We used two NASH animal models: the first was a rat model, whose liver exhibits characteristic NASH histopathology, similar to that of human NASH, including steatosis, fibrosis, lipid droplets, and hepatocyte ballooning (Figure 1D). The second was a mouse model, also presenting characteristic NASH pathology. A recent study showed that the mouse model developed HCC when bred for long enough [28], making it ideal for the study of hepatic precancerous lesions. However, while almost all NASH patients are obese, these models exhibited weight loss. No weight loss was reported in a recently described hamster model of NASH [20], and we intend to confirm our findings using this model.

We analyzed changes in carbohydrate metabolism during NASH using the two animal models, and proteome analysis indicated that glycolysis was activated (Figure 2B). Western blotting revealed that PKM2 was significantly upregulated in NASH, as was p-PKM2, its active form (Figure 2C,D). IHC further indicated that PKM2 was specifically upregulated in Kupffer cells (Figure 5D). These results suggested that glycolysis is promoted in Kupffer cells during NASH. Kupffer cells, which are macrophages present in hepatic sinusoids, have phagocytic activity against pathogens entering via portal circulation [29]. Moreover, Kupffer cells produce cytokines, including proinflammatory IL-6, IL-1β, IFN-γ, and TNF-α [30].

It has been reported that lipopolysaccharides activate Kupffer cells, which then release cytokines such as TNF-α, promoting steatohepatitis in NASH [31,32]. Macrophages are broadly classified into classically activated M1 and alternatively activated M2 [33]. M1 macrophages are predominantly expressed in NASH [34,35], and glycolysis is significantly upregulated in M1 macrophages due to inflammation [36,37]. Further, M1 macrophages release more proinflammatory cytokines, such as TNF-α, IL-6, and IL-12, than M2 [38]. These cytokines stimulate hepatic stellate cells and promote liver fibrosis in NASH [39]. On the other hand, in M2 macrophages, oxidative phosphorylation is more pronounced than glycolysis, and they release anti-inflammatory cytokines, such as IL-10, promoting repair after inflammation [38]. Interestingly, during carcinogenesis, Kupffer cells re-polarize from M1 to a specific M2 phenotype, known as tumor-associated macrophages (TAMs) [40,41]. In the present study, increased Kupffer cell and glycolysis activation was observed in NASH (Figure 2B). These data indirectly suggested an upregulation of M1 Kupffer cells. In addition, the increase in PKM2 and p-PKM2 levels during the early phase of NASH indicated M1 polarization and phagocytic properties (Figure 4). In addition, previous reports have shown that the transformation of TAM, the M2 phenotype, to the M1 phenotype suppressed carcinogenesis in pancreatic cancer [42,43], as well as metabolic reprogramming by directly targeting PKM2 altered the phenotype of Kupffer cells [44]. Inferring from our study results and these reports, the transformation of Kupffer cell phenotype pertaining to glycolysis shows promise in liver fibrosis and HCC treatment. Therefore, changes in the glycolytic metabolism of TAMs warrant further investigation. Furthermore, the pyruvate dehydrogenase complex (PDC) catalyzes the oxidation of pyruvate to produce acetyl CoA. Previous studies have reported that PDC activity is reduced in NAFLD, and several NAFLD therapies target PDC reactivation [45,46,47]. These study results corroborate our present study, which focused on the glycolytic system in NASH/NAFLD.

We also explored the mechanism underlying PKM2 upregulation, focusing on miR-122-5p, the most abundant miRNA within the liver [15,16]. We confirmed that miR-122-5p downregulated PKM2 expression, while the levels of miR-122-5p were significantly lower in Kupffer cells during NASH (Figure 5). Several previous studies have indicated that miR-122-5p directly binds to the 3′UTR of *PKM* and suppresses PKM2 expression in HCC [48,49]. Therefore, we considered miR-122-5p downregulation as one factor that promotes glycolysis in Kupffer cells during NASH, as a precancerous phenomenon. We previously reported a strong relationship between miRNA regulation and the Warburg effect in cancer cells [10,50]. The current results also support this connection; that is, the Warburg effect is closely regulated by miRNAs, and miRNA dysregulation profoundly contributes to early-phase carcinogenesis. However, there have been no reports of miR-122-5p regulating PKLR expression, and the binding of miR-122-5p and 3’UTR of PKLR was not confirmed in the database (TargetScan7.2). Although a replacement of PKM2 with PKLR was observed as the disease progresses to NASH via immunohistochemistry and Western blotting (Figure 4), the actual mechanism of regulation of PKLR and PKM2 expression levels was presumed to be different. Additionally, an additional experiment that we performed showed that the expression of PKLR was reduced in the Kupffer cell line (KUP5) treated with miR-122-5p (data not shown). Further experiments are needed to elucidate this phenomenon. Although this report does not prove that inflammatory cytokine expression changes miR-122-5p-transfected Kupffer cells, a previous study has reported that miR-122-5p inhibited the release of inflammatory cytokines in hepatic stellate cells [51] and it is presumed that miR-122-5p also regulates cytokinesis in Kupffer cells.

Our study has some limitations. First, we used only two diet-based animal models, which showed weight loss and decreased triglyceride levels (data not shown). Therefore, we must consider whether these models accurately reflect the pathophysiology of NASH. There are several genetically modified NASH models, such as *ob*/*ob* mice, *db*/*db* mice, and Pten^-/-^ mice [52,53,54]. However, these only harbor a single gene mutation that does not recapitulate the pathophysiological mechanism of NASH in patients. Furthermore, considering the multiple hits hypothesis of NASH [55] proposed in recent years, diet-based models exhibit extensive alterations that resemble NASH in patients to a greater extent than in genetic models. Second, this study did not directly show that M1 Kupffer cells, rather than M2, are increased. Hence, further experiments are required to assess M1/M2 qualitative changes. Macrophage markers such as CD80, CD86, CD163, and CD206 may be utilized to test our hypothesis. Third, the cause of the miR-122-5p downregulation observed in our study remains unknown.

Studies have shown that methylation of CpG islands in the miR-122 promoter region reduces the expression of miR-122-5p in HCC and liver injury [56]. Other reports have demonstrated that liver-enriched transcription factors, such as HNF1α, HNF3β, and HNF4α, regulate the expression of miR-122-5p in HCC [57,58,59]. Thus, these factors may also coordinate miR-122-5p expression in Kupffer cells during NASH. Accordingly, we intend to further investigate the mechanisms underlying miR-122-5p regulation.

In conclusion, we identified miR-122-5p as an essential factor that regulates glycolysis and the Warburg effect in Kupffer cells at a precancerous stage during NASH.

## 4. Materials and Methods

### 4.1. Animal Models

Ten-week-old male SHRSP5/Dmcr rats were obtained from Japan SLC (Shizuoka, Japan) and randomly divided into two groups. The control group was fed an SD, while the NASH group was fed an HFC diet (Funabashi Farms Co., Ltd., Chiba, Japan; each *n* = 4). Six-week-old male C57BL/6J mice were also obtained from Japan SLC and randomly divided into two groups. The control group was fed SD, while the NASH group was fed a CDAHFD (A06071302, Research Diets, New Brunswick, NJ, USA; each *n* = 5). The animals were housed in a temperature of 25 °C and 12 h light/dark-controlled room with ad libitum access to tap water. Plasma was separated from blood samples via centrifugation at 3000× *g* for 15 min at 4 °C. Plasma levels of AST and ALT were measured by SRL Inc. (Tokyo, Japan). All procedures involving animals were conducted according to the guidelines of Osaka Medical and Pharmaceutical University (Approval No. 2020-094) and the Guide for the Care and Use of Laboratory Animals by the National Academy of Sciences.

### 4.2. Histological Analysis and Immunohistochemistry (IHC)

Hepatic tissue specimens were fixed with Carnoy’s fixative in 10% methanol overnight. Fixed liver tissues were embedded in paraffin and cut to a thickness of 4 µm. The sections were mounted on adhesive glass slides (Matsunami Glass Ind., Osaka, Japan) and deparaffinized with xylene and ethanol. Hepatic histological changes were assessed using HE staining and Sirius Red staining (Polysciences, Inc., Warrington, PA, USA). For IHC staining, endogenous peroxidase activity and non-specific antigens were blocked with 10% H_2_O_2_ and Serum-Free Ready-to-Use Protein Block (Dako, Santa Clara, CA, USA), respectively, followed by incubation of the sections with antibodies against PKM2, F4/80 (Cell signaling Technology, Inc., Danvers, MA, USA), PKLR, and CD68 (Abcam, Cambridge, MA, USA) overnight at 4 °C. Sections were then washed with PBS and incubated with EnVision Dual Link System-HRP (Dako) as the secondary antibodies at room temperature for 30 min. Immunoreactions were visualized using 3,3-diaminobenzidine solution (Nichirei Biosciences Inc., Tokyo, Japan) and counterstained with hematoxylin. Finally, the sections were dehydrated and mounted. Images were taken with a BZ-x700 microscope (KEYENCE, Tokyo, Japan). The antibodies used are described in Appendix A.

### 4.3. Sample Preparation for Phosphoproteomics

Liver tissues were homogenized into powder using a Cryoprep device (CP02; Covaris, Woburn, MA, USA). Each tissue was transferred to a Covaris tissue bag (TT1; Covaris) and pulverized at an impact level of 3. To enhance the identification of proteins in the hydrophobic samples, lysis buffer (4% sodium deoxycholate (SDC), 0.1 M Tris-HCl pH 8.5, and one phosphatase inhibitor tablet in 10 mL) was mixed with the samples using SDC [60,61]. Additional sonication was performed using a focused ultrasonicator, and the lysate was centrifuged at 16,000× *g* for 10 min. The supernatant was collected and the debris was further lysed using a probe sonicator and then centrifuged at 16,000× *g* for 10 min. Additional supernatant was combined with the previous supernatant, and the protein concentration was measured via a BCA (bicinchoninic acid) protein assay (BCA Protein Assay Kit; Pierce, Waltham, MA, USA). Two milligrams of each tissue lysate was divided among four tubes (500 µg each) and digested via filter-aided sample preparation [62]. The prepared protein was reduced using a sodium dodecyl sulfate buffer (4% SDC in 0.1 M Tris-HCl, pH 7.6, and 0.1M DTT) for 45 min in a 37 °C incubator. The reduced proteins were sonicated for 10 min in a bath sonicator (5800; Branson, Danbury, CT, USA). Samples were transferred to 30 kDa Microcon centrifugal filters (Merck Millipore, Burlington, MA, USA) and centrifuged in a filter (16,000× *g*, 40 min, 20 °C). The buffer was exchanged with 200 µL of 8 M urea and 0.1 M Tris pH 8.5, and this step of SDC removal was repeated three times. To prevent re-bonding of the cysteine bond, alkylation using 0.05 M iodoacetamide (IAA; Sigma-Aldrich, St. Louis, MO, U.S.A.) was performed. The 8 M urea and 0.1 M Tris pH 8.5 buffer was changed three times, and the 50 mM triethylammonium bicarbonate (TEAB; pH 8) buffer was changed two times. Trypsin was added to the sample at a 1:50 ratio for digestion, and the sample was incubated at 37 °C overnight.

The digested peptides were desalted using a C18 spin column. For comparative analysis, desalted peptides were labeled using the 10-plex TMT reagent as per the manufacturer’s instructions [63]. NASH peptides were labeled as 126, 127N, 128N, 129N, and 130N, and control peptides were labeled as 127C, 128C, 129C, 130C, and 131 TMT reagents. After labeling the TMT reagents, each labeled peptide was pooled and concentrated using vacuum centrifugation. Mid-pH reverse-phase liquid chromatography fractionation was adopted for our experiment. The TMT-labeled peptides were fractionated into 12 fractions. An analytical column (Xbridge, C18 5 µm, 4.6 mm × 250 mm) coupled with an Agilent 1260 series HPLC system (Agilent, Santa Clara, CA, USA) was used to separate peptides. The gradient was as follows: 0–10 min, 5% B; 10–70 min, 35% B; 70–80 min, 70% B; 80–105 min, 5% B; 10 mM TEAB in water, pH 7.5 in mobile phase A; 10 mM TEAB in 90% ACN, pH 7.5 in mobile phase B. Each fraction was collected and dried using vacuum centrifugation. Immobilized metal affinity chromatography was used for phosphopeptide enrichment. The Ni-NTA magnetic bead (36113; Qiagen GmbH, Venlo, Netherlands) slurry was washed three times using deionized water (DW) and then reacted with 100 mM ethylenediaminetetraacetic acid (EDTA; pH 8.0) for 30 min. After the Ni^2+^ ions were removed, the EDTA solution was removed and the beads were washed three times with DW. The beads were then reacted with 10 mM aqueous FeCl_3_ solution for 30 min. The Fe^3+^-NTA beads were washed three times with DW and resuspended in 1:1:1 ACN/MeOH/0.01% acetic acid. Phosphopeptides from each fraction were enriched using a 12-tube magnet (36912; Qiagen GmbH). The Fe^3+^-NTA beads in each tube were washed with 400 μL binding buffer (80% ACN/0.1% TFA). The TMT-labeled peptide was resuspended in 500 μL of binding buffer and transferred to a tube containing the aliquoted beads. While binding reactions proceeded for 30 min, end-over-end rotation was performed. Washing was performed four times (using a 500 μL binding buffer). Finally, the phosphopeptides were enriched via incubation in 125 μL of 1:1 ACN/2.5% ammonia in 2 mM phosphate buffer (pH 10) for 1.5 min. The enriched phosphopeptides were acidified with 10% trifluoroacetic acid.

### 4.4. LC-MS and Phosphoproteomics Analysis

Phosphopeptides were resuspended in solvent A (0.1% formic acid in water), injected into an Q Exactive (Thermo Fisher Scientific, Waltham, MA, USA), and coupled to an EASY-nLC 1000 instrument (Thermo Fisher Scientific). A 180 min gradient was used as follows: 5% solvent B equilibration for 10 min, 5% to 40% solvent B for 130 min, 40% to 80% solvent B for 10 min, holding at 80% solvent B for 15 min, and equilibrating the column for 5% solvent B for 15 min. A trap column (2 cm × 75 µm ID packed with 2 µm C18 resin) and an analytical column (50 cm × 75 µm ID packed with 2 µm C18 resin) were used to fractionate the peptides. To identify as many peptides as possible, the data-dependent acquisition method was adopted, and the top 10 precursor ions were selected and isolated in ± 0.8 *m*/*z* windows for fragmentation. MS1 data were acquired with a mass range of 400–2000 *m*/*z* at a resolution of 70,000. The dynamic exclusion time was set as 30 s, and the normalized collision energy was set as 30. MS2 scans were acquired at a resolution of 17,500 with the first 100 *m*/*z* fixed. The maximum ion injection time was 50 ms, and the automated gain control target value was set to 1.0 × 10^6^. Raw data were processed with post-experiment monoisotopic mass refinement to increase sensitivity during peptide identification [64].

The SEQUEST HT search option in Proteome-Discoverer 2.0 (Thermo Fisher Scientific) was used for protein identification using preprocessed data. The Swiss-Prot rat database (released in July 2019) was consulted. More than two missed cleavages were filtered out for accurate peptide identification. Carbamidomethylation, alkylation of S-bonds in cysteine, TMT 6-plex modification of lysine, and N-termination were noted as static modifications, while methionine oxidation and phosphorylation in serine, threonine, and tyrosine were noted as variable modifications. The false discovery rate for peptides was set to 0.01 (high confidence in both peptide and protein levels) to remove false-positive data. Phosphorylation site localization was validated using phosphoRS [65]. To enrich differentially expressed proteins, fold changes in expression between NASH and controls (>1.5 and <0.67) with p-values below 0.05 from the results of Student’s *t*-tests were selected as strict cut-off values. Gene Ontology analysis was performed using the database for annotation, visualization, and integrated discovery in order to interpret protein and phosphoprotein function [66]. An interaction network was constructed using Cytoscape and the Kyoto Encyclopedia of Genes and Genomes pathway database [67,68].

### 4.5. Western Blotting

Liver tissues were homogenized in chilled lysis buffer comprising 10 mM Tris-HCl (pH 7.4), 1%NP-40, 0.1% deoxycholic acid, 0.1% sodium dodecyl sulfate, and 150 mM NaCl (Thermo Fisher Scientific), followed by incubation for 20 min on ice. After centrifugation at 12,000× *g* for 20 min at 4 °C, the supernatants were collected as protein samples. Protein content was determined using a DC™ Protein Assay kit (Bio-Rad, Hercules, CA, USA). Seven micrograms of protein lysate were separated via SDS-PAGE using 10% polyacrylamide gels (FUJIFILM Wako, Osaka, Japan) and electroblotted onto a PVDF membrane (Bio-Rad). After blocking non-specific binding sites for 1 h with 5% non-fat milk (Cell Signaling Technology, Inc.) in PBS containing 0.05% Tween 20 (PBS-T), the membrane was incubated with primary antibodies overnight at 4 °C. On the following day, the membrane was washed three times with PBS-T, incubated with secondary antibodies at room temperature (20–25 °C) for 1 h, and washed three times with PBS-T. The immunoblots were visualized using Immobilon Forte Western HRP Substrate (Merck Millipore). The protein bands were detected using FUSION-FX7 (Vilber Lourmat, Marne-la-Vallée, France) [69]. The antibodies that were used are described in Appendix A.

### 4.6. Quantitative Real-Time Reverse Transcription-PCR (qRT-PCR)

The miRNA was isolated from tissues using a NucleoSpin microRNA isolation kit (Takara Bio Inc., Shiga, Japan) according to the manufacturer’s protocols. To determine the expression levels of miR-122-5p, we conducted qRT-PCR using TaqMan MicroRNA Assays (Applied Biosystems, Foster City, CA, USA) and THUNDERBIRD Probe qPCR Mix (TOYOBO Co., LTD., Osaka, Japan) according to the manufacturer’s protocols. Relative expression levels were calculated via the 2^–∆∆Ct^ method. Each value of ∆∆Ct was determined using the Thermal Cycler Dice Real-Time System II model TP870 (Takara Bio Inc.). *SNO202* and *RNU6B* were used as internal controls for the mice and rats, respectively.

### 4.7. Cell Culture

The HCC cell line HuH-7 was obtained from the Japanese Collection of Research Bioresources Cell Bank. The Kupffer cell line KUP5 was obtained from the RIKEM BioResource Research Center [70]. HuH-7 was cultured in DMEM (FUJIFILM Wako) supplemented with 10% (*v*/*v*) heat-inactivated fetal bovine serum (FBS). KUP5 was cultured in DMEM with 10% (*v*/*v*) FBS, 10 µg/mL bovine insulin, and 250 µM monothioglycerol. Both cell lines were cultured at 37 °C in a humidified atmosphere containing 5% CO_2_.

### 4.8. Transfection Experiments

The HuH-7 cells were seeded in 6-well plates at a concentration of 0.5 × 10^5^ cells per well (10–30% confluence) on the day before transfection. Each transfection was performed using Lipofectamine™ RNAiMAX (Invitrogen, Carlsbad, CA, USA) as per manufacturer instructions. Cells were transfected with mature miR-122-5p (mirVana™ miRNA mimic; Thermo Fisher) and antagomiR-122-5p (mirVana™ miRNA inhibitor; Thermo Fisher). The sequence of the mature miR-122-5p used in this study was 5′-UGGAGUGUGACAAUGGUGUUUG-3′, while that of the non-specific miRNA (HSS, Hokkaido, Japan) was 5′-GUAGGAGUAGUGAAAGGCC-3′.

### 4.9. Luciferase Reporter Assay

To identify the miR-122-5p binding sites, we used the Target Scan 7.2 database (http://www.targetscan.org/, accessed on 31 March, 2022) and found the predicted binding site to be at position 520–527 within the 3′UTR of PKM mRNA. The sequence region containing the putative binding sequence was inserted into a pMIR-REPORT^TM^ Luciferase miRNA Expression Reporter Vector (Applied Biosystems Inc.) according to the manufacturer’s protocol. We also generated other pMIR constructs, including a mutated seed sequence for miR-122-5p (wild-type, ACACUCC; mutant, ACUGACC), using a PrimeSTAR^®^ Mutagenesis Basal Kit (Takara Bio Inc.) [23]. The mutation of each vector was confirmed via sequence analysis. A pRL-TK *Renilla* Luciferase Reporter vector (Promega Corporation, Madison, WI, USA) was used as an internal control vector. Cells were seeded into 96-well plates at a concentration of 0.1 × 10^4^ per well on the day before transfection. Each cell type was co-transfected with a reporter vector (wild-type or mutant; 0.01 µg/well) and miR-122-5p or a non-specific non-coding siRNA (Dharmacon, Tokyo, Japan). Luciferase activity was measured using a Dual-Glo Luciferase Assay System (Promega Corporation) and reported as the Firefly luciferase/*Renilla* luciferase ratio [23].

### 4.10. Isolation of Kupffer Cells

The Kupffer cells were obtained from mouse livers after 12 weeks of feeding with the HFC diet. Mouse liver tissues were perfused with Hanks’ Balanced Salt Solution (Thermo Fisher Scientific) and Collagenase type I solution (Merck Millipore). Tissue samples were filtered through 100 µm filter. Samples were then subjected to low-speed centrifugation (500× *g* for 3 min, repeated three times), and non-parenchymal cells (NPCs) were separated from hepatocytes and isolated from the supernatant. To isolate Kupffer cells, NPCs were centrifuged in 50%/25% Percoll (GE Healthcare, Waukesha, WI, USA). The Kupffer-cell-enriched layer was plated and cultured in DMEM supplemented with 10% FBS at 37 °C in a humidified atmosphere containing 5% CO_2_ for 30 min [71].

### 4.11. Statistics

All statistical analyses were performed in triplicate. The two-tailed Student’s *t*-test and Mann–Whitney *U* test were used to analyze differences for in vitro and in vivo experiments, respectively. Data are presented as means ± standard deviation (SD). Statistical significance was set at *p* < 0.05. Graphs were created using GraphPad Prism version 7.00 for Windows (GraphPad Software, San Diego, CA, USA).

## Figures and Tables

**Figure 1 ijms-23-05230-f001:**
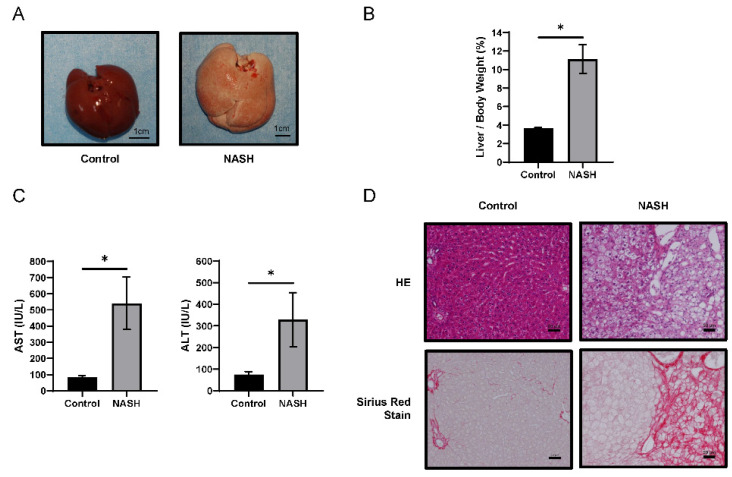
Comparison between SHRSP5/Dmcr rats fed a standard diet (SD) or high-fat and high-cholesterol (HFC) diet for 8 weeks. (**A**) Representative macroscopic appearance of control (*n* = 4) and non-alcoholic steatohepatitis (NASH) (*n* = 4) livers. Scale bars = 1 cm. (**B**) The liver-to-body weight ratio in each group. (**C**) Aspartate transaminase (AST) and alanine transaminase (ALT) levels in plasma in each group. Values represent mean ± SD. Note: * *p <* 0.05 vs. control. (**D**) Hematoxylin–eosin- and Sirius Red-stained liver sections. Scale bars = 50 µm.

**Figure 2 ijms-23-05230-f002:**
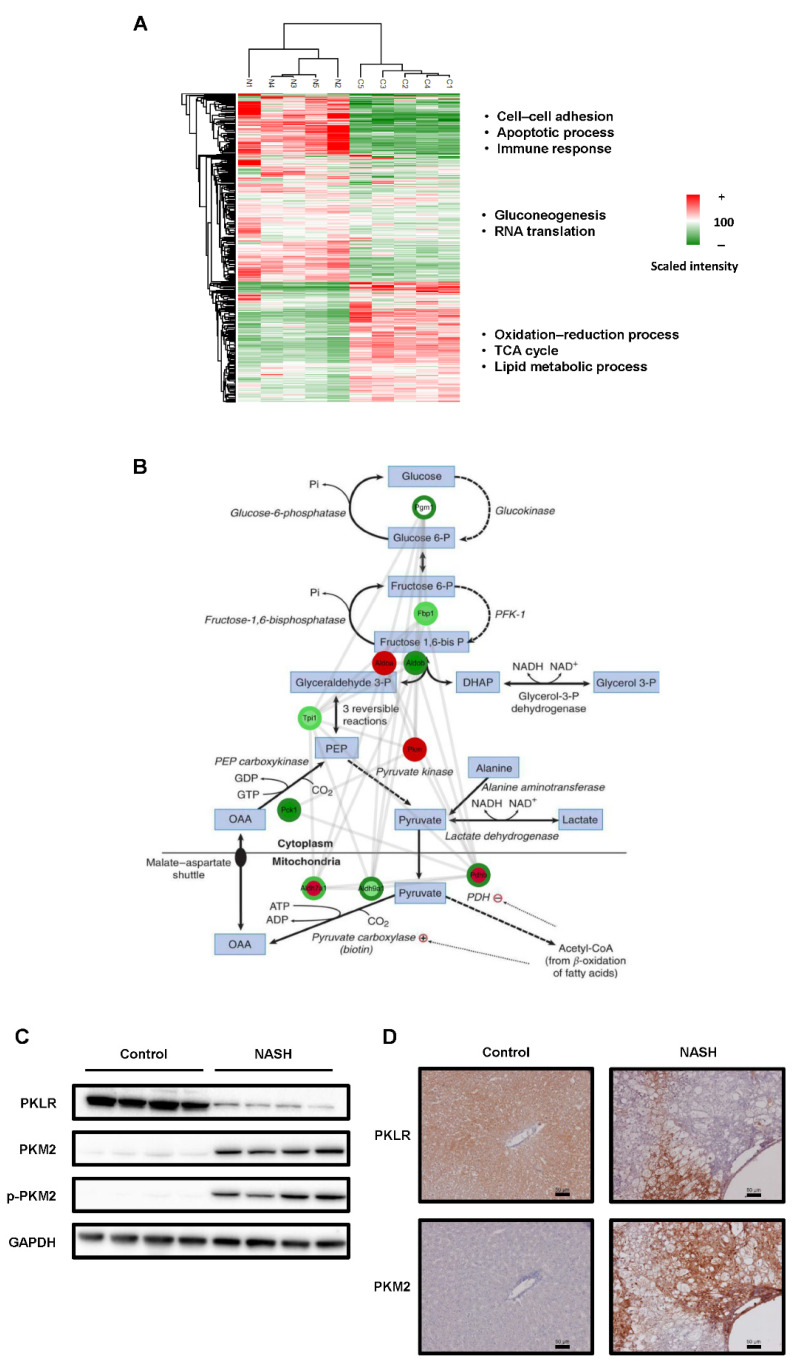
Changes in glycolysis and the expression of rate-limiting enzyme PKM2 in rat models of non-alcoholic steatohepatitis (NASH). (**A**) Heat map of the phosphoproteome. Quantitative changes in protein phosphorylation of each pathway. A significant difference was observed for each pathway. (**B**) Phosphoproteome analysis of the glycolytic pathway. Note: inner circle, global protein; outer circle, phosphoprotein. Note: ADP, Adenosine diphosphate; ATP, Adenosine triphosphate; DHAP, Dihydroxyacetone phosphate; GDP, Guanosine diphosphate; GTP, Guanosine triphosphate; NAD, Nicotinamide adenine dinucleotide; OAA, Oxaloacetic acid; PDH, Pyruvate dehydrogenase; PEP, Phosphoenolpyruvic acid. (**C**) Protein expression levels of PKLR, PKM2, and phospho-PKM2 in liver samples from each group. GAPDH was used as an internal control. (**D**) Immunohistochemistry (IHC) of each group’s liver samples. Scale bars = 50 µm.

**Figure 3 ijms-23-05230-f003:**
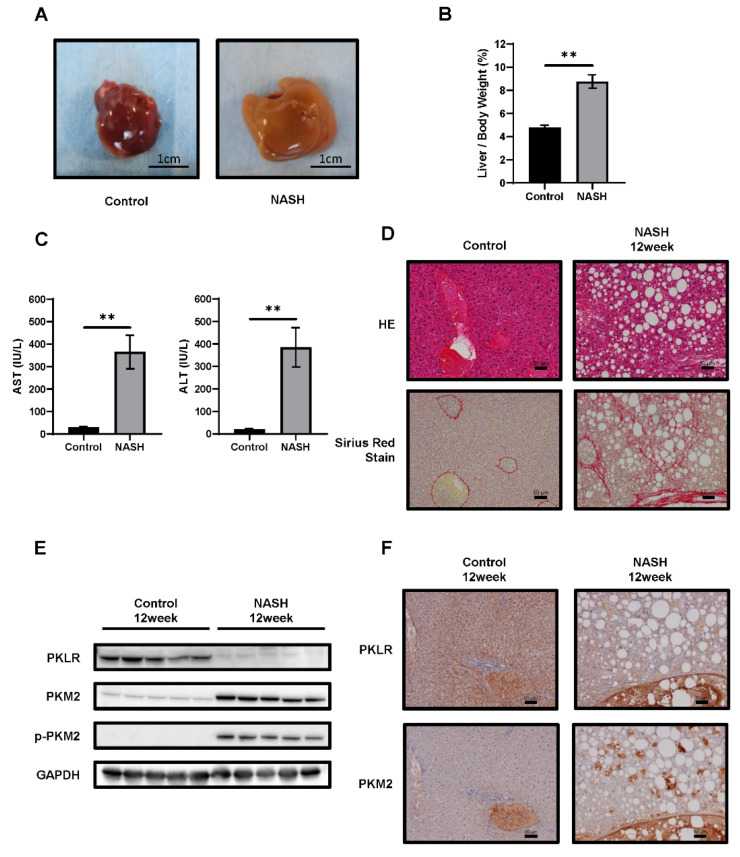
Comparison between C57BL/6J mice fed the standard diet (SD) or choline-deficient, L-amino-acid-defined, high-fat diet (CDAHFD) for 12 weeks. (**A**) Representative macroscopic appearance of control (*n* = 5) and non-alcoholic steatohepatitis (NASH) (*n* = 5) livers. Scale bars = 1 cm. (**B**) The liver-to-body weight ratio in each group. (**C**) AST and ALT levels in plasma. Values represent the mean ± SD. Note: ** *p* ˂ 0.01 vs. control. (**D**) HE-stained and Sirius Red-stained liver sections. Scale bars = 50 µm. (**E**) Western blotting analysis and (**F**) immunohistochemistry (IHC) of the expression levels of PKM2, phosphorylated PKM2 (p-PKM2), and PKLR. GAPDH was used as an internal control. Scale bars = 50 µm.

**Figure 4 ijms-23-05230-f004:**
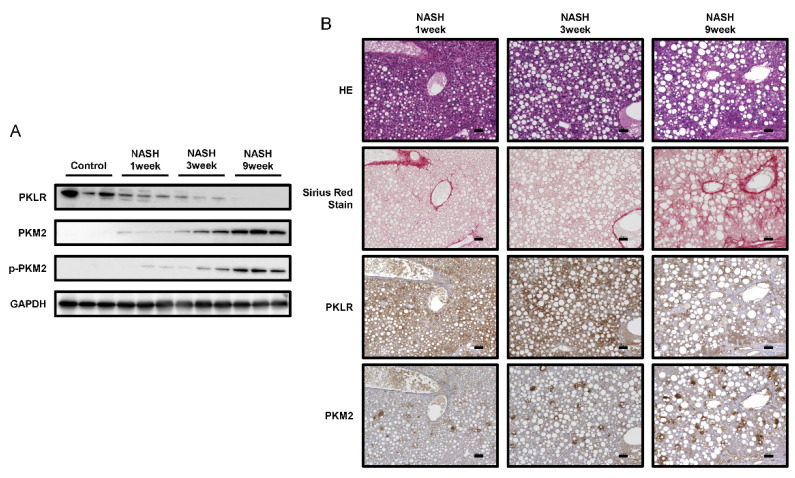
Changes in PK enzyme expression in the early-phase non-alcoholic steatohepatitis (NASH) mouse model. (**A**) Protein expression levels of PKLR, PKM2, and phosphorylated PKM2 (p-PKM2) in the liver of control and NASH mice after 1, 3, and 9 weeks (*n* = 3). GAPDH was used as an internal control. (**B**) Images of each staining. Upper panels, HE; 2nd panels, Sirius Red staining; 3rd panels, PKLR; lower panels, PKM2. Scale bars = 50 µm.

**Figure 5 ijms-23-05230-f005:**
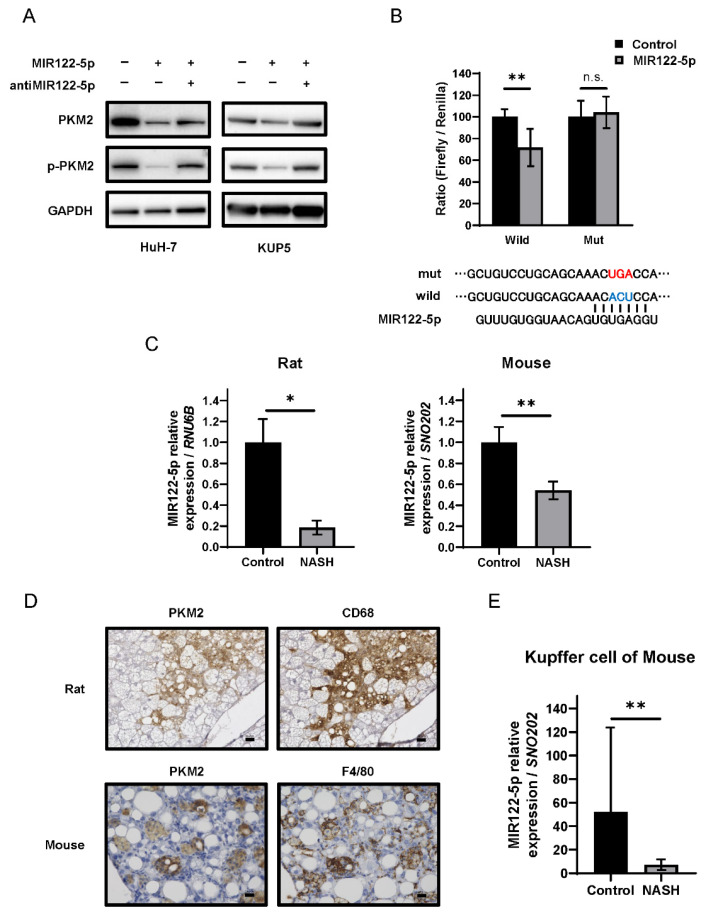
Association between miR-122-5p and PKM2 expression in Kupffer cells. (**A**) PKM2 and p-PKM2 expression under treatment with miR-122-5p or miR-122-5p inhibitor. The effect was examined in the HCC cell line (HuH-7) and Kupffer cell line (KUP5). Left, non-specific miRNA (Control; 20nM); middle, miR-122-5p mimic (20nM); right, miR-122-5p mimic and inhibitor (each 10nM). The effects were assessed at 72 h after treatment. GAPDH was used as an internal control. (**B**) The region of the 3’UTR of *PKM* mRNA complementary to the mature miR-122-5p. The lower segment indicates the predicted binding sites for miR-122-5p. Luciferase activities after co-transfection with control, miR-122-5p, and wild-type or mutant pMIR vectors harboring the predicted miR-122-5p binding site in the 3′UTR of *PKM*. Note: Red text, mutant type; Blue text, wild type. (**C**) The expression of miR-122-5p in each non-alcoholic steatohepatitis (NASH) model. *RNU6B* and *SNO202* were used as internal controls for rats and mice, respectively. (**D**) Immunohistochemistry (IHC) of PKM2 and Kupffer cell markers (CD68 for rats or F4/80 for mice). Scale bars = 50 µm. (**E**) MiR-122-5p expression in Kupffer cells. SNO202 was used as a control. Values represent the mean ± SD. Note: * *p* ˂ 0.05, ** *p* ˂ 0.01, and n.s., not statistically significant vs. control.

**Figure 6 ijms-23-05230-f006:**
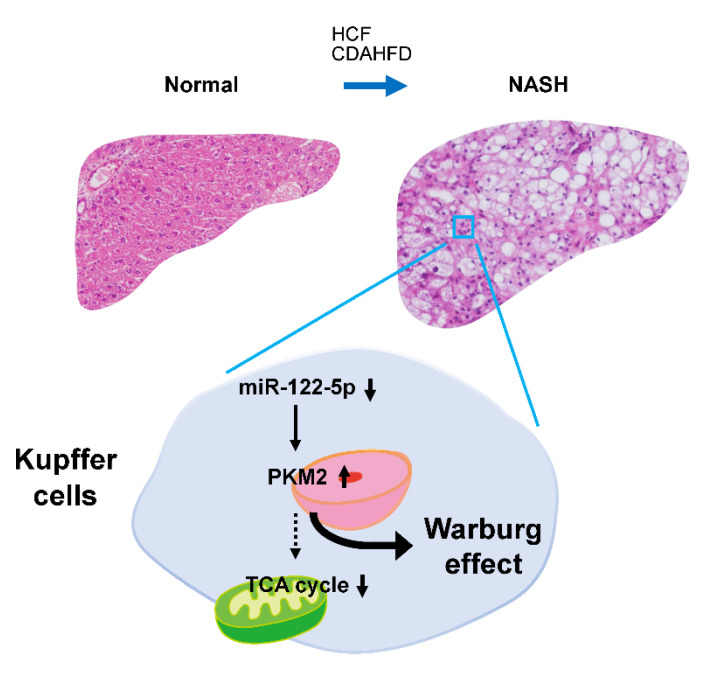
Schematic diagram of the miR-122-5p-mediated regulation of the Warburg effect in Kupffer cells during non-alcoholic steatohepatitis (NASH). The Warburg effect, particularly glycolysis, was activated in animal models of NASH. One of the main mechanisms underlying this activation was the upregulation of PKM2. Notably, dysregulation of miR-122-5p partially induced PKM2 upregulation, especially in Kupffer cells during NASH. These findings suggest that miRNA is strongly related to metabolic changes in carcinogenesis, even during the precancerous phase.

**Table 1 ijms-23-05230-t001:** Validations of microRNA (miRNA) used in this study.

miRNA gene name (ID)	miR-122 (406906)
Target gene name (ID)	PKM (5315)
Species name (ID)	Homo sapiens
Genomic location of MTI Nucleotide sequence	5′-ACACTCCA
Location	15:72199548-72199555
Location within a part of a gen	520-527 (location within 3’UTR)
Existence of precious report (searching with miRTarBase)	Yes
Methods for experimental validation in this study	Luciferase reporter assay Western blot analysis
Experimental materials used in this study	HuH-7 and KUP5 cell lines

Note: 3′UTR, 3′ untranslated region; HuH-7, human hepatocellular carcinoma cell lines; KUP5, mouse Kupffer cell lines; MIR, microRNA (gene symbol); MTI, microRNA–target interaction; PKM, pyruvate kinase M1/2.

## Data Availability

All relevant data are contained within this article and in the Appendix A.

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
