# Peer review of "Downregulation of miR-122-5p Activates Glycolysis via PKM2 in Kupffer Cells of Rat and Mouse Models of Non-Alcoholic Steatohepatitis"

_ijms, 2022, doi:10.3390/ijms23095230_

Round 1

Reviewer 1 Report

The authors explored the role of miR-122-5p in two animal models with NASH. They found downregulation of miR-122-5p was correlated with dysregulation of PKLR/PKM, which resulted in the activation of glycolysis in Kuppfer cells. The results were further confirmed by cell studies. The experiment design is good, and the results can support the conclusion. However, some issues are still needed to be addressed.

  1. The significance of the finding is needed to be further clarified. Upregulation of glycolysis can be observed in macrophage during carcinogenesis. However, does “normalization” of the metabolic pathway prevent hepatocellular carcinoma? A further discussion is needed.
  2. NASH results in liver cirrhosis and HCC, which are both important health issues. In the discussion, the author only focused on the possible impact of miR-122-5p on HCC. Will inhibition of glycolysis in Kupffer cells benefit liver fibrosis in NASH? If this is difficult to answer in the current study, at least a discussion is warranted.
  3. Is it possible to check the expression of cytokines in KUP5 with or without miR122-5p modulation? This may further support the correlation between this pathway and inflammation.
  4. PKLR and PKM2 are both isoforms of pyruvate kinase. In the rats, PKM2 loci were replaced by PKLR foci. However, this was not so prominent in mice. Would it possible to check the expression of PKLR in cell experiments?

Reviewer 2 Report

The present study points to interesting findings in the identification of miR-122-5p as an essential factor in the regulation of glycolysis in Kupffer cells during the progression of NASH. However, there are some points to be rethought:

  1. Major changes:
  2. The grading of histopathological alterations contributes to the characterization of fatty liver disease. Semi-quantitative measures, such as the NAFLD activity score, as proposed by Kleiner et al. (PMID: 15915461) are useful to characterize and differentiate the development of simple steatosis from nonalcoholic steatohepatitis, cirrhosis, etc. I suggest these analyses be added to the study.
  3. Previous results in the literature have shown that pharmacological reactivation of the pyruvate dehydrogenase (PDH) complex prevents the worsening of nonalcoholic fatty liver disease. I suggest that these studies be added to the introduction or discussion to reinforce the study hypothesis (PMID: 35032488; PMID: 33860188 and PMID: 29656110).
  4. Based on the findings of the present study, I suggest that the discussion be rethought as the authors focus on a simplified view of the M1/M2 cell profile and inflammatory markers and they did not make any measurement of inflammatory process. If the aim of the study was to assess miRNA regulation of glycolysis in NASH models, the focus of the discussion should be the same.
  5. Were there observed changes in diet intake? Having that information is relevant for this type of study.
  6. Minor changes:
  7. Review the legends of the figures. I suggest that information related to the description of results should be restricted to the results. Example: Figure 4, line 159-160: “This change became more pronounced over time”.
  8. Correct some writing errors. Example: page 3, line 87: “SHERSP5/Dmcr”.

Reviewer 3 Report

Thank you for this fundamental research article. This study provides relevant data to the patho-physiology of NASH and its precancerous state which is currently not fully elucidated. The  miR-122-5p is nowadays extensively studied but not in this medical area.  The methods are thor-ough and fine. The Discussion is well balanced, concise and include all the current data available on this subject. For that purpose this research article could be accepted in present form.

Round 2

Reviewer 1 Report

The author had addressed all of my questions.